# Soil Actinomycetes of Vietnam Tropical Forests

Yuliya A. Dorchenkova [1,*], Tatiana A. Gracheva [1], Tamara L. Babich [2], Diyana Sh. Sokolova [2], Alina V. Alexandrova [3,4], Giang T. H. Pham [4], Lyudmila V. Lysak [1], Alla V. Golovchenko [1] and Natalia A. Manucharova [1]

[1] Soil Science Department, Lomonosov Moscow State University, Leninskie Gory 1-12, Moscow 119991, Russia
[2] Winogradsky Institute of Microbiology, Research Center of Biotechnology, Russian Academy of Sciences, Leninsky Prospect 33-2, Moscow 119071, Russia
[3] Department of Biology, Lomonosov Moscow State University, Leninskie Gory 1-12, Moscow 119991, Russia
[4] Joint Vietnam-Russia Tropical Science and Technology Research Center, 63 Nguyen Van Huyen, Nghia Do, Cau Giau, Hanoi 100000, Vietnam
* Correspondence: juliadorchenkova@gmail.com

**Abstract:** Actinomycetes are an important group of bioactive hydrolytic bacteria in any ecosystem. However, the actinomycete biodiversity in tropical ecosystems, particularly in Vietnam, is still underexplored. The aim of this article is to analyze the abundance, taxonomic structure and ecophysiological features of actinomycete complexes of soils and litter in Vietnam's protected areas. A total of 41 samples of soils, plant litter and suspended soils were collected from six of Vietnam's national parks and nature reserves. The direct inoculation technique showed that the total abundance of actinomycetes varied from $2.0 \times 10^4$ to $1.0 \times 10^8$ CFU/g. According to the luminescent microscopy with acridine orange dye, the length of the actinomycete mycelium was as long as 1000 m/g in the litter of Xuan Son National Park. A total of 80 strains were isolated and tested for antagonistic activity against *Bacillus subtilis*, *Aspergillus niger* and *Candida albicans*. Inoculation on Getchinson's medium showed high cellulolytic activity. The most active strains were isolated from alluvial brown soil, plant litter and suspended soil of the Pu Hoat Nature Reserve. In these samples, actinomycetes adapted to high temperatures and low pH were found to be predominant. High-throughput sequencing of the V3–V4 region of the 16S rRNA gene and bioinformatic analysis confirmed the high taxonomic diversity and high hydrolytic activity of actinomycete complexes of the Pu Hoat Nature Reserve samples.

**Keywords:** *Actinobacteria*; tropical forests; ferralsols; biodiversity; antibiotic activity; high-throughput sequencing; 16S rRNA gene; cellulose degradation

## 1. Introduction

The study of the biological diversity of various regions of our planet is an issue of paramount importance in view of the increasing anthropogenic impact on natural biogeocenoses. The richest areas in terms of species diversity are the tropical forests of North and South America, as well as Southeast Asia [1]. The study of the abundance and diversity of microbial communities in soils under tropical forests is important both in terms of understanding the role of microorganisms in soil formation processes and in terms of their participation in the cycle of biophilic elements under tropical climate conditions. The biodiversity data and knowledge about the functioning of ecosystems in the humid subtropics are of great importance for the protection and rational use of the region's resources.

The phylum *Actinobacteriota* currently includes six classes: *Acidimicrobiia*, *Actinomycetia*, *Coriobacteriia*, *Nitriliruptoria*, *Rubrobacteria* and *Thermoleophilia* [2–10]. Of the greatest biotechnological importance among them is the class *Actinomycetia*. These are Gram-positive bacteria with a high G + C content, most of which have the ability to form branching

mycelium. Mycelial actinobacteria are widely distributed in nature. They are found in the air, in fresh and salt water, in food, intestines and excrements of invertebrates, but their greatest diversity is found in soil and plant substrates [11]. Such a variety of habitats for actinomycetes is due to their high resistance to adverse conditions, such as temporary lack of nutrients or dry seasons. Actinomycetes are found in all types of soils, including the northernmost and southernmost bioclimatic zones [12–15]. Due to the surge of interest of microbiologists in actinomycetes with unique antibiotic properties, the importance of studying natural sources of these microorganisms from extreme habitats has recently increased. The acidic soils that form under tropical forests are one of these little-studied sources [16–19].

Forest soils of a boreal climate with an acidic pH are characterized by a low content of actinomycetes compared to soils with a neutral or slightly alkaline pH, where they are decomposers of organic matter [20,21]. It has also been found that a characteristic feature of the prokaryotic complex of acidic tropical soils is the high abundance and wide taxonomic diversity of mycelial actinobacteria [22–25], significantly distinguishing these complexes from the acidic soils of the boreal climate.

The property of actinomycetes that is the most interesting for scientists is their ability to synthesize antibiotics. Most antibiotics in medical practice are natural or semi-synthetic compounds derived from actinomycetes and fungi. It is worth noting that due to the growing problem of antibiotic resistance of pathogenic microorganisms, the search for new antibiotics and strains that produce them becomes more and more urgent. Analysis of the genome sequences of some *Streptomyces* showed that they are capable of producing 10 times more active metabolites than previously known [26]. For this reason, actinomycetes remain a promising source of new biologically active substances. Studying the patterns of distribution of actinomycetes is of interest not only in connection with the search for producers of biologically active compounds, but also from the point of view of resistome formation, especially in warm climate soils [27].

## 2. Materials and Methods

### 2.1. Materials

The material was collected during the expedition of the Vietnam-Russian Tropical Research and Technological Centre from 2016 to 2018 in Xuan Son National Park, Phu Tho Province; Pu Hoat Nature Reserve and Pu Mat National Park, Nghe An Province; Kon Plong Protected Forest, Kon Tum province; and Kon Chu Rang Nature Reserve and Kon Ka Kinh National Park, Gia Lai Province. All samples were taken once at the beginning of the wet season. Samples were taken according to the method generally accepted for soil microbiological studies, uniformly in all the studied parks and reserves. Sample plots of 400 sq·m (20 × 20 m) were established for sampling in typical park landscapes. For each site, a description was made, and then 10 point samples were randomly selected. From these samples, a thoroughly mixed composite sample was made in the laboratory. Samples were collected according to the following scheme: plant litter, soil at a depth of 0–5 cm and 5–20 cm (Table 1). In the forests of Pu Hoat Nature Reserve, Kon Ka Kinh National Park and Kon Chu Rang Nature Reserve, we collected suspended soils forming in the baskets of epiphytes of the genera *Asplenium* (*Aspleniaceae*) and *Drynaria* (*Polypodiaceae*).

### 2.1.1. Xuan Son National Park

The national park is located in the Phu Tho province in the northern part of Vietnam; its area is about 15 thousand hectares. The climate is characterized by hot summers and cool winters, with an average annual temperature of 22–23 °C. The studied substrates in this territory were collected at 2 sites. Site 1 is a tropical low-mountain polydominant broadleaf karst forest with a predominance of representatives of the families *Elaeocarpaceae*, *Lauraceae*, *Moraceae*, *Sabiaceae* and *Anacardiaceae* on karst rocks in the valley of a temporary watercourse (21.121648° N, 104.945771° E, 580 m a.s.l.). The soil is dark brown ferrallitic on limestone. The litter is abundant, forming a thick layer consisting of leaves at various

stages of decomposition (Figure 1). Site 2 is located on the Mount Ten peak; it is a tropical mid-mountain polydominant broadleaf forest on granite with a predominance of representatives of the families *Fabaceae*, *Lauraceae* and *Magnoliaceae* (21.115354° N, 104.934785° E, 1200 m a.s.l.). The soil is mountainous dark brown ferrallitic, covered with litter from banana trees and fallen trunks lying on the ground. The litter is abundant, forming a thick layer, represented by leaves at different stages of decomposition.

**Table 1.** Ecotopes (sorted from north to south) and soil types according to the World Reference Base for Soil Resources (WRB) in sampling sites.

| Protected Area | Sampling Location (Altitude, m) | Soil Type (WRB) |
|---|---|---|
| Xuan Son National Park | Karst valley, broadleaf forest (580 m) | Calcisols |
| | Mount Ten, broadleaf forest (1200 m) | Ferralsols |
| Pu Hoat Nature Reserve | Broadleaf forest in river valley (845 m) | Fluvisols |
| | | Suspended soils |
| | Tall-stemmed coniferous forest (1370 m) | Ferralsols |
| Pu Mat National Park | Tall-stemmed forest in river valley (200 m) | Ferralsols |
| Kon Chu Rang Nature Reserve | Broadleaf forest (1000 m) | Ferralsols |
| | Coniferous forest (1050 m) | Ferralsols |
| | Broadleaf forest (1025 m) | Ferralsols |
| | | Suspended soils |
| | Mixed forest (995 m) | Ferralsols |
| Kon Plong protected forest | Tall-stemmed mixed forest (1400 m) | Ferralsols |
| Kon Ka Kinh National Park | Eastern part of the park, forest on slope (700 m) | Cambisols |
| | | Suspended soils |
| | Forest on ridge (900 m) | Ferralsols |
| | Southwestern part of the park, tall-stemmed forest in valley (1000 m) | Ferralsols (Gleyic) |
| | Southwestern part of the park, mid-mountain tall-stemmed forest (1500 m) | Ferralsols |
| | Southwestern part of the park, low-stemmed forest (860 m) | Fluvisols |
| | Southwestern part of the park, mixed forest (1160 m) | Cambisols |

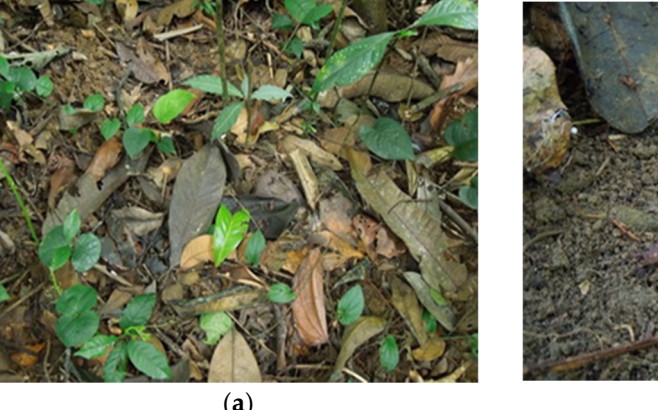
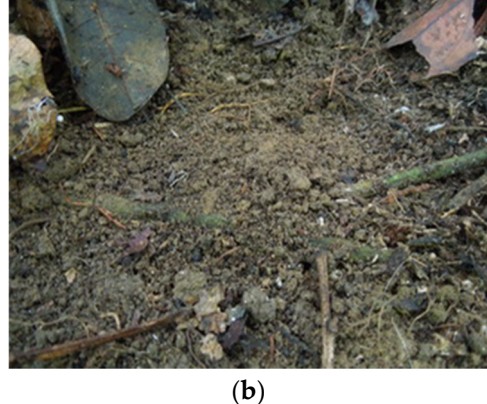

(**a**)  (**b**)

**Figure 1.** Leaf litter (**a**) and dark brown ferrallitic soil (**b**) sampled from Site 1 of the Xuan Son National Park.

### 2.1.2. Pu Hoat Nature Reserve

The Pu Hoat Nature Reserve is located in the Nghe An province. In this territory, the samples were collected at 2 sites: in a river valley and on a ridge slope. Site 1 is a tropical valley broadleaf polydominant tall-stemmed forest with a predominance of *Terminalia* sp. and *Aglaiagigantean* trees with a vertical structure of medium complexity on waterlogged alluvial soils on granite in the valley of the Zut Suoi River (19.762038° N, 104.802386° E, 845 m a.s.l.). The soil is alluvial brown with fragments of granite. Partially fragmented leaf litter with a thickness of 3–5 cm is at different stages of decomposition of plant material. At this site, we collected suspended soils from baskets of epiphytes of the *Aspleniaceae* and *Polypodiaceae* family [28]. Site 2 is a tropical tall-stemmed mountain forest with a predominance of *Cunninghamia lanceolate* (*Cupressaceae*) trees on granite on a steep ridge slope (19.775998° N, 104.803729° E, 1370 m a.s.l.). The soil is mountainous red-yellow humus–ferrallitic. Coniferous litter from twigs and needles of *Cunninghamia* with a thickness of 10–15 cm is fragmented at different stages of decomposition of plant material.

### 2.1.3. Pu Mat National Park

Pu Mat National Park is located in the northern part of Central Vietnam, in the Nghe An province; it occupies 194 thousand hectares. The samples were collected at a site in a tropical valley polydominant tall-stemmed permanently humid forest with the presence of *Dracontomelondao*, *Bischofia javanica*, etc., in the valley of the Khe Choang River on its shales (18.955816° N, 104.685032° E, 200 m a.s.l.). The soil is red-yellow ferrallitic. Leaf litter with a thickness of 3–5 cm is partially fragmented at different stages of decomposition of plant material.

### 2.1.4. Kon Chu Rang Nature Reserve

Kon Chu Rang Nature Reserve is located on a mountain plateau in the Gia Lai province in Central Vietnam (14°26′–14°35′ N, 108°30′–108°39′ E); its area is 15,900 hectares. In the territory of this reserve, sampling was carried out at 4 sites. Site 1 is a tropical low-mountain broadleaf polydominant tall-stemmed forest with a predominance of trees from the families *Lauraceae*, *Burseraceae*, *Myrtaceae* and *Hamamelidaceae* on a permanently wet gentle slope on fragmented basalts (14.51795° N, 108.54593° E, 1000 m a.s.l.). The soil is mountainous red-yellow humus–ferrallitic. The litter with a thickness of 1–2 cm does not completely cover the soil surface; it consists of partially fragmented leaves at different stages of decomposition (Figure 2). Site 2 is a tropical low-mountain mixed forest with a predominance of trees from the family *Podocarpaceae* (*Dacrydium elatum*), as well as less-represented *Hamamelidaceae* (*Simingtonia*), *Rhodoliaceae* (*Rhodolia*), *Fagaceae* and *Sterculiaceae* (*Scaphium*) with a vertical structure of medium complexity on short-profile soils on a basalt slab (14.48856° N, 108.56924° E, 1050 m a.s.l.). The soil is mountainous red-yellow humus–ferrallitic. The litter covers the soil surface in a thick layer of 10–15 cm; it consists of twigs and needles of *Dacrydium elatum* with individual leaves at different stages of decomposition. Site 3 is a tropical low-mountain broadleaf forest with a predominance of trees from the families *Dipterocarpaceae*, *Clusiaceae*, *Ebenaceae*, *Fabaceae*, etc., on the edge of the plateau on short-profile soil on basalt (14.514043° N, 108.571246° E, 1025 m a.s.l.). The soil is mountainous red-yellow humus–ferrallitic. Leaf litter of a thickness of 1–2 cm is partially fragmented with plant material at different stages of decomposition. In this territory, we collected suspended soil from the baskets of epiphytes of the *Dryopteridaceae* family. Site 4 is a low-mountain mixed forest with a predominance of trees from the *Podocarpaceae* family, as well as less-represented *Hamamelidaceae*, *Rhodoleiaceae*, *Myrtaceae*, etc., with a vertical structure of medium complexity on waterlogged short-profile soils on a basalt slab (14.468823° N, 108.562208° E, 995 m a.s.l.). The soil is red-yellow humus–ferrallitic. Deciduous–coniferous litter consisting of needles and twigs of *Dacrydium elatum* with a thickness of 10–15 cm is partially fragmented with plant material at different stages of decomposition.

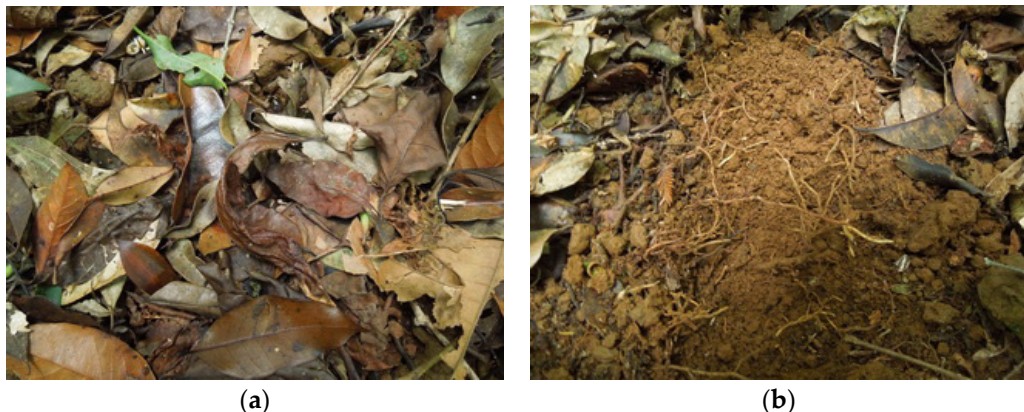

(**a**)                (**b**)

**Figure 2.** Leaf litter (**a**) and mountainous red-yellow humus–ferrallitic soil (**b**) sampled from Site 1 of the Kon Chu Rang Nature Reserve.

2.1.5. Kon Plong Protected Forest

The Kon Plong Protected Forest located in the Kon Tum province of Central Vietnam is characterized by a tropical monsoon climate with a wet season from May to October and a dry season from November to April. Soil and litter samples were collected at a site in a tropical mid-mountain mixed polydominant tall-stemmed permanently humid forest with a predominance of trees from the families *Podocarpaceae* (*Dacrycarpus imbricatus*), *Magnoliaceae* (*Michelia, Mangletia, Kmeria*), *Myrtaceae (Syzygium)*, *Calophyllaceae* (*Calophyllum*), *Elaeocarpaceae* (*Slonea*) and *Betulaceae* (*Betula*) on a gentle slope with outcrops of granite fragments (14.753985° N, 108.297858° E, 1400 m a.s.l.). The soil is mountainous red-yellow humus–ferrallitic. The litter with a thickness of 1–2 cm does not completely cover the soil surface; it consists of partially fragmented leaves at different stages of decomposition.

2.1.6. Kon Ka Kinh National Park

Kon Ka Kinh National Park is located on the Kon Tum Plateau in the Gia Lai province of Central Vietnam (14°09′–14°30′ N, 108°16′–108°28′ E); its area is 41,780 hectares. The terrain is mountainous, with altitudes varying from 570 m in the river valleys to 1748 m at the top of the Kon Ka Kinh mountain. The area has a contrasting, tropical monsoon climate with a wet season from May to November and a dry season from December to April. In this territory, samples were collected at 6 sites, the first two in the eastern part of the park, the rest in the southwestern. Site 1 is a forest on a slope (14.296336° N, 108.445607° E, 700 m a.s.l.) in the eastern part of the Kon Ka Kinh National Park. The soil is brown tropical thin sandy loam. Leaf litter is partially fragmented at different stages of decomposition. Sampling was carried out on suspended soil from epiphyte baskets. Site 2 is a forest on a ridge (14.320337° N, 108.444608° E, 900 m a.s.l.) in the eastern part of the national park. The soil is mountainous red-yellow humus–ferrallitic. Leaf litter is represented by individual leaves and sprigs of needles. Site 3 is a tropical low-mountain valley polydominant permanently wet tall-stemmed forest with a predominance of trees from the families *Euphorbiaceae, Myrtaceae, Moraceae, Duabangaceae, Lauraceae, Fagaceae* and *Meliaceae* in the valley of the A Yun River in the southwestern part of the national park (14.21937° N, 108.31765° E, 1000 m a.s.l.). The soil is hydromorphic dark humus–ferrallitic with traces of gleying. The litter lies in a thick layer, represented by leaves at different stages of decomposition that have fallen at different times. Site 4 is a tropical mid-mountain polydominant tall-stemmed forest with a predominance of trees from the families *Juglandaceae, Fagaceae, Elaeocarpaceae* and *Magnoliaceae* on a wide crest of the ridge (14.22287° N, 108.331880° E, 1500 m a.s.l.). The soil is mountainous red-yellow humus–ferrallitic. The litter completely covers the soil surface in a layer of 2–5 cm (1–2 leaves); it is very dry and consists of partially fragmented leaves at different stages of decomposition. Site 5 is a light low-stemmed valley forest in the southwestern part of the Kon Ka Kinh National Park (14.217081° N, 108.283478° E, 860 m a.s.l.). It is a tropical flooded polydominant forest on light soils, with a predominance of *Shorea siamensis* Miq.,

*Shorea roxburghii* G. Don (*Dipterocarpaceae*), *Schima* (*Theaceae*), *Irvingia* (*Irvingiaceae*), *Streblus asper* Lour. *Ficus* spp. (*Moraceae*) and *Syzygium* (*Myrtaceae*). It is located in a wide flooded river valley and remains waterlogged for a long time. The soil is sandy loamy alluvial on shales and clays. The litter consists of large leaves, partially washed away by water flows. Site 6 is a mixed forest in the southwestern part of the national park (14.193672° N, 108.323651° E, 1160 m a.s.l.). It is a tropical mid-mountain tall-stemmed forest with a predominance of *Pinus dalatensis* (*Pinaceae*), *Elaeocarpus* (*Elaeocarpaceae*), *Schima* (*Theaceae*), *Podocarpus neriifolius* (*Podocarpaceae*) and *Rhodoleia* (*Hamamelidaceae*). It grows in a wide ridge and on a gentle slope of a ridge composed of slates. The soil is brown forest and well-drained. The litter forms a thick, dense layer (15–20 cm), composed of leaves, needles and twigs, differentiated into layers according to the degree of decomposition.

### 2.2. Research Methods

Soil pH was determined in the filtered supernatant of water suspensions of soil samples (soil:water = 1:2.5). To determine the acidity of the litter, we used the filtrate of an equilibrium solution prepared in the proportion of litter:water = 1:25.

The total abundance of actinomycetes in the substrates was determined by inoculation of suspensions at serial dilutions of 1:1000 and 1:10,000 on the Gause-1 solid nutrient medium (mineral agar 1) (g/L): $K_2HPO_4$—0.5; $MgSO_4$—0.5; $KNO_3$—1; $NaCl$—0.5; $FeSO_4$—traces; starch—20; agar—20; pH 7.2–7.4 [29]. To prepare a suspension, 1 g of soil or litter was taken and placed in a flask with 100 mL of sterile water. To desorb mycelium from soil particles, the suspension was treated with a Bandelin Sonopuls HD 2070 ultrasonic disperser (Germany) for 2 min at a power of 50%. Each dilution was cultured in three replications. Plates were incubated in a thermostat at 28 °C for 7–10 days, and then the total number of grown colonies was counted. The number of cultivated actinomycetes per 1 g of soil (CFU/g) was calculated according to the formula $CFU = \frac{a*n*b}{C}$, where a is the average number of colonies per dish; n is the dilution from which the inoculation was made; b is the volume of plated drop of suspension, mL; and c is the soil sample, g.

The length of the actinomycete mycelium was measured on a luminescent microscope by staining the preparations of the soil suspension with acridine orange dye [30]. We used the same suspensions as for the inoculation method. For one sample, 6 preparations were made on two defatted glass slides. Then, 0.01 mL of the suspension was applied to each preparation with a micropipette and distributed over an area of 4 cm$^2$. Then, the preparation was dried in air, fixed over a burner flame and stained with an aqueous solution of acridine orange (the working dye solution was used at a concentration of 1:10,000). The exposure time of preparations in the dye was 3 min, then rinsing in tap water was 2 times for 5 min. After staining, the preparations were dried at room temperature and examined under a LUMAM-IZ microscope (Russia). Light filters ZhS-19, ZhS-18, objective lens (×90 L) and eyepieces (×4 or ×5) were used. On each preparation, in 50 fields of view, the length of fragments of actinomycete mycelium was measured using an ocular ruler. Based on all measurements, the average length of the actinomycete mycelium in the field of view was determined. The length of actinomycete mycelium in 1 g of soil (N, m/g) was calculated by the formula: $N = \frac{S1*a*n}{v*S2*c} * 10^6$, where S1 is the preparation area, μm$^2$; a is the average length of actinomycete mycelium in the field of view, μm; n is the suspension dilution index, mL; v is the volume of a drop applied to glass, mL; S2 is the microscope field of view area, μm$^2$; and c is the soil sample, g.

Pure cultures of actinomycetes were isolated by successive subculturing using Gause-1 medium (mineral agar 1); inoculations were incubated at 28 °C.

The antibiotic activity of actinomycete strains was determined by the agar block method, incubating the inoculations for 20–24 h at a temperature favorable for the development of test organisms (*Bacillus subtilis*, *Aspergillus niger* and *Candida albicans*) [31]. Joint cultivation of test organisms and actinomycetes was carried out on glucose–peptone–yeast agar (g/L) [29]: glucose—1; peptone—2; yeast extract—1; casein hydrolyzate—1; $CaCO_3$—1; agar—20; glycerin—10 mL; pH 7.2.

The ability to degrade cellulose in the isolated strains of actinomycetes was identified by the presence of growth on Getchinson's medium with filter paper (g/L): $K_2HPO_4$—1; $CaCl_2$—0.1; $MgSO_4$—0.3; NaCl—0.1; $FeCl_3$—traces; $NaNO_3$—2.5; agar—20; pH 7.2.

To determine the temperature range and optimal growth temperature, pure cultures of actinomycetes were cultivated at temperatures of 4, 10, 15, 22, 30, 37, 41, 45, 49.5 and 54 °C for 5 days in a liquid PC medium (g/L): NaCl—0.5; tryptone—5; yeast extract—2.5; glucose—1. The optical density of strains in the liquid medium was measured at a wavelength of $\lambda = 660$ nm ($OD_{660}$).

To identify acid-tolerant and acidophilic strains of actinomycetes, pure cultures were sown on Gause-1 solid media with pH values of 5.0 and 4.0. The media were prepared using a phosphate–citrate buffer mixture (0.2 M $Na_2HPO_4$, 0.1 M citric acid).

DNA isolation, amplification and sequencing of the 16S rRNA gene. Identification of pure cultures of actinomycetes was carried out by sequencing of the 16S rRNA gene. The isolated DNA of pure cultures was PCR-amplified with primers that are universal for representatives of the *Bacteria* phyla: 8-27f [5′-AGAGTTTGATCCTGGCTCAG-3′] and 1492r [5′-GGTTACCTTGTTACGACTT-3′]. PCR was carried out in a reaction mixture (25 μL) containing 10–50 ng of DNA template using an iCycler thermal cycler (BioRad, Hercules, CA, USA) in the following mode: 3 min at 94 °C, followed by 30 cycles (0.5 min at 94 °C, 0.5 min at 50 °C, 0.5 min at 72 °C), then 7 min at 72 °C. The length of the resulting fragments was assessed on a 1.0% agarose gel with ethidium bromide. DNA sequencing was performed using the ABI PRISM® BigDye™ Terminator v. 3.1 Kit and an ABI 3730 DNA Analyzer (Applied Biosystems, Foster City, CA, USA) in accordance with the manufacturer's recommendations. Preliminary analysis of the resulting nucleotide sequences was performed using the BLAST search against the NCBI GenBank database. The resulting sequences were compared with those of reference-type organisms. Sequence editing was performed using the BioEdit program [32,33].

High-throughput sequencing of 16S rRNA gene fragments. To assess the taxonomic diversity of actinomycete complexes, we used the method of high-throughput pyrosequencing of the variable region of the 16S rRNA gene in total soil DNA. The method is based on the identification of an evolutionarily conserved gene. During the study, an analysis of the V3–V4 hypervariable region of the 16S rRNA gene was performed for each of the microorganisms in the sample. The study was carried out by the next generation sequencing (NGS) method using an Illumina MiSeq platform with subsequent bioinformatics processing of the obtained data. Sample preparation was carried out using the two-stage polymerase chain reaction (PCR) technique. In the first stage, amplification of the hypervariable V3–V4 region of the 16S rRNA gene was performed using the primers universal for all prokaryotes. In the second stage, the PCR product obtained in the first stage was amplified with the purpose of barcoding the library. The resulting amplicons after purification on magnetic particles and measurement of concentration by the fluorometric method were ready-made DNA libraries suitable for multiplex sequencing on the Illumina platform. Subsequently, DNA analysis was performed on a new generation Illumina MiSeq sequencer using the paired end reads (2 × 300 bp), generating at least 10,000 paired reads per sample. Processing of sequencing data was carried out using the QIIME 1.9.1 automated algorithm, which includes combining forward and reverse reads, removing technical sequences, filtering sequences with low reliability of reads for individual nucleotides (quality less than Q20), filtering chimeric sequences, aligning reads to the 16S rRNA reference sequence and distributing sequences by taxonomic units using the Silva database version 132. The classification algorithm of operational taxonomic units (OTUs) with an open reference (Open Reference OTU picking) was used; the classification threshold was 97%.

Bioinformatics analysis. The functional characteristics of bacterial communities were predicted using the Local Mapper module of the iVikodak software package [34] and the KEGG database [35]. Heat maps were built using the ClustVis Internet resource (http://biit.cs.ut.ee/clustvis accessed on 11 July 2022). The Venn diagram was built using the Venny 2.1 online resource (https://bioinfogp.cnb.csic.es/tools/venny/ accessed on 14

July 2022). The sequences of the 16S rRNA gene of microbial communities were deposited to GenBank under the accession number PRJNA861932.

## 3. Results

### 3.1. Determination of the Total Abundance of Actinomycetes, the Length of the Actinomycete Mycelium and the Structure of the Actinomycete Complex in the Studied Samples

The reaction of the medium in the samples of soils and plant litter was weakly acidic to acidic, and a high content of representatives of the *Actinobacteriota* phylum was noted in all substrates. The data obtained are presented in Table 2.

In the territory of the Xuan Son National Park, two types of soils were analyzed: dark brown ferrallitic and mountainous dark brown ferrallitic. In the flatland soil and substrates associated with it, the abundance of actinomycetes was higher than in the mountainous soil, reaching the maximum value of $1.0 \times 10^8$ CFU/g in the litter; the greatest length of the actinomycete mycelium was noted there as well, 1000 m/g. In the mountainous soil, a high content of mycelium was also observed; however, unlike the flatland soil, its content in the litter was two times lower. The taxonomic structure of the actinomycete community was dominated by *Albus albus* and *Cinereus achromogenes* sections and series (Figure 3).

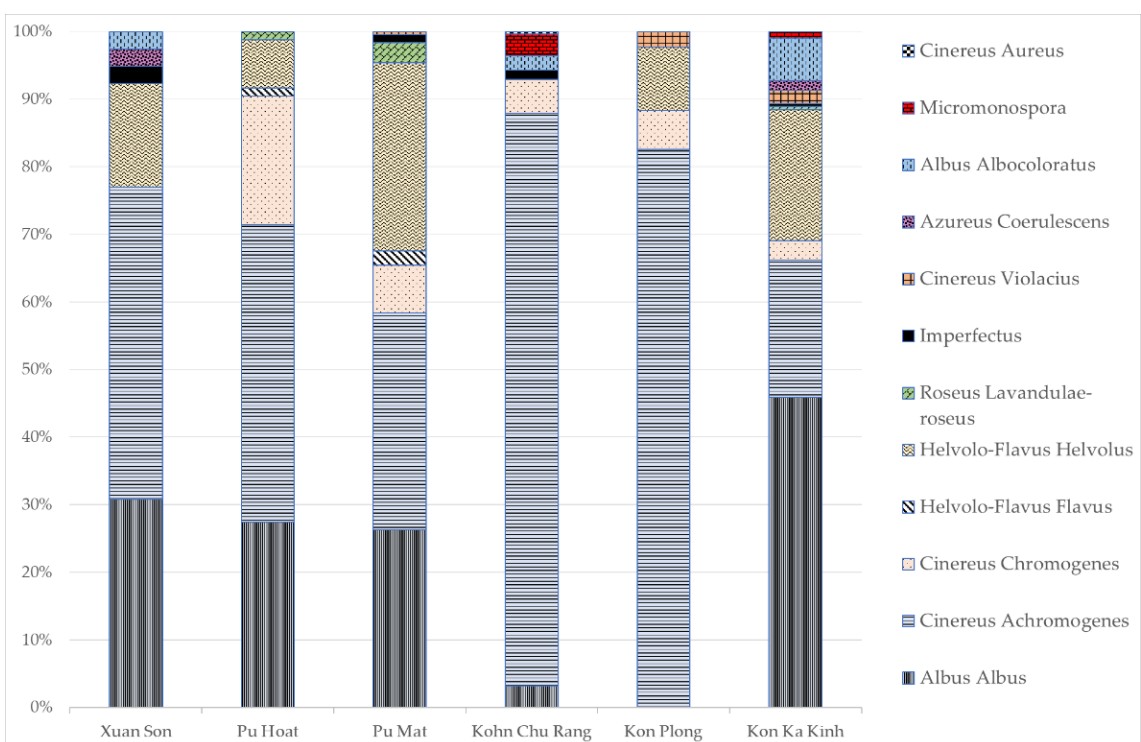

**Figure 3.** The taxonomic composition of actinomycete complexes at the section and series level based on morphological characteristics of cultured actinomycetes. The figure shows percentage content of the actinomycete section (Y-axis) in the actinomycete complex of protected area (X-axis).

In the Pu Hoat Nature Reserve, which is located south of Xuan Son, two types of soils were analyzed: alluvial brown and mountainous red-yellow humus–ferrallitic. The highest abundance (10 million CFU/g) was noted for the litter formed above the soil located in lowland areas. In the same territory, we studied samples of suspended soil collected from the baskets of epiphytic plants. The abundance of actinomycetes in them reached $6.0 \times 10^5$ CFU/g. At the same time, a similar content and distribution pattern of mycelium along the profile was noted for both mountainous and alluvial soils: the maximum content of mycelium was observed in the litter; it sharply decreased in the upper soil layer and increased again at a depth of 20 cm. The taxonomic structure of the actinomycete complex of the Pu Hoat Nature Reserve included streptomycetes of *Albus albus*, *Cinereus chromogenes*,

*Cinereus achromogenes*, *Helvolo-Flavus helvolus*, *Helvolo-Flavus flavus* and *Roseus lavandulae-roseus* sections and series. Dominants among them were actinomycetes of sections with gray and white aerial mycelium.

In the Pu Mat National Park, two horizons of red-yellow humus–ferrallitic soil and leaf litter were analyzed. The distribution pattern of actinomycetes in the studied substrates of the red-yellow humus–ferrallitic soil was similar to that for the Pu Hoat Nature Reserve. The greatest length of actinomycete mycelium (800 m/g) was noted in the litter; it sharply decreased in the upper soil layer and increased again almost twofold at a depth of 20 cm. The structure of the actinomycete complex of the Pu Mat National Park was characterized by a greater diversity. In addition to streptomycetes, we identified actinomycetes of the genus *Micromonospora*. Actinomycetes of *Cinereus achromogenes*, *Helvolo-Flavus helvolus* and *Albus albus* sections and series were predominant.

In the Kon Chu Rang Nature Reserve located in the southern part of Vietnam, red-yellow humus–ferrallitic soil was studied at four sites: a mountain plateau, in a relief depression, under a broadleaf forest and under a coniferous forest. The soil located in a relief depression was distinguished by a higher abundance of actinomycetes and a higher mycelium content compared to the mountainous soil. At the same time, in the litter above the ferrallitic soil under the coniferous forest, the length of the actinomycete mycelium was several times greater than in the litter under the broadleaf forest. The structure of the actinomycete complex was characterized by a relatively high diversity; it was represented by *Cinereus chromogenes*, *Cinereus achromogenes*, *Cinereus aureus*, *Albus albus*, *Albus albocoloratus* and *Imperfectus* sections and series. Representatives of the *Micromonospora* genus were also distinguished, and actinomycetes with gray aerial mycelium were predominant.

In the territory of the Kon Plong Protected Forest in Central Vietnam, horizons of mountainous red-yellow humus–ferrallitic soil and leaf litter were also analyzed. The largest abundance of actinomycetes (1 million CFU/g) was observed in the litter formed over mountainous red-yellow humus–ferrallitic soil, which confirms the role of actinomycetes as the main decomposers of organic matter in the microbial community of plant litter. The length of actinomycete mycelium varied from 240 to 400 m/g of the substrate. The diversity of actinomycetes was relatively low; the community was mainly represented by the *Cinereus* section. At the same time, the vast majority of actinomycetes in the soil were represented by the *Cinereus achromogenes* section and series.

In the southern part of Vietnam in the territory of the Kon Ka Kinh National Park, we analyzed soils sampled at six different sites from the eastern and southwestern parts of the park. In general, this park is characterized by a high abundance of actinomycetes and a low content of mycelium in the substrates, indicating the predominant existence of actinomycetes in the form of spores. In the eastern part of the park, the abundance of actinomycetes was higher in the brown tropical soil located in relatively low relief conditions. In this territory, we also analyzed samples of suspended soil from epiphyte baskets. The abundance of actinomycetes in it reached $10^7$ CFU/g, while mycelium was not detected. In the southwestern part, the presence of actinomycetes in the form of mycelium was noted in the alluvial brown soil under hydromorphic conditions. In other substrates, a high abundance of actinomycetes ($10^6$ CFU/g) indicated that actinomycetes are present in the form of spores, which is probably due to sampling carried out during the dry climatic period. The studied substrates of this national park located to the south of all the others are characterized by the richest taxonomic diversity of streptomycetes.

Thus, actinomycetes are widespread in the studied tropical biotopes. The abundance of actinomycetes, mycelium length and species diversity in the studied ecosystems was high. In actinomycete complexes, the most prevalent were representatives of the *Cinereus achromogenes* section and series, while the second place in prevalence was occupied by streptomycetes of *Albus albus*, *Helvolo-Flavus helvolus* and *Cinereus chromogenes* sections and series.

**Table 2.** The number of cultivated actinomycetes, CFU/g (average value according to the inoculation method) and length of actinomycete mycelium (m/g) in the studied substrates.

| Protected Area | Ecotope | Sample | Abundance of Actinomycetes, CFU/g | Length of Actinomycete Mycelium, m/g |
|---|---|---|---|---|
| Xuan Son National Park | Karst valley, broadleaf forest, dark brown ferrallitic soil | Litter | $1.0 \times 10^8$ | 1000 |
| | | Soil (0–20 cm) | $0.5 \times 10^6$ | 300 |
| | Mount Ten, broadleaf forest, mountainous dark brown ferrallitic soil | Litter | $2.0 \times 10^5$ | 400 |
| | | Soil (0–20 cm) | $0.6 \times 10^5$ | 800 |
| Pu Hoat Nature Reserve | River valley, broadleaf forest, alluvial brown soil | Litter | $9.2 \times 10^6$ | 800 |
| | | Soil (0–5 cm) | $9.3 \times 10^5$ | 178 |
| | | Soil (5–20 cm) | $3.1 \times 10^5$ | 406 |
| | River valley, broadleaf forest, suspended soil | Soil from epiphyte baskets | $6.0 \times 10^5$ | 580 |
| | Ridge slope, tall-stemmed forest, mountainous red-yellow humus–ferrallitic soil | Litter | $1.1 \times 10^5$ | 800 |
| | | Soil (0– 5 cm) | $2.0 \times 10^5$ | 200 |
| | | Soil (5–20 cm) | $2.0 \times 10^4$ | 406 |
| Pu Mat National Park | River valley, tall-stemmed forest, red-yellow humus–ferrallitic soil | Litter | $1.4 \times 10^5$ | 820 |
| | | Soil (0–5 cm) | $8.5 \times 10^5$ | 240 |
| | | Soil (5–20 cm) | $1.1 \times 10^6$ | 380 |
| Kon Chu Rang Nature Reserve | Broadleaf forest, mountainous red-yellow humus–ferrallitic soil | Litter | $2.0 \times 10^4$ | 200 |
| | | Soil (0–5 cm) | $2.0 \times 10^4$ | 280 |
| | | Soil (5–20 cm) | $4.1 \times 10^5$ | 360 |
| | Broadleaf forest, suspended soil | Soil from epiphyte baskets | $2.1 \times 10^5$ | 680 |
| | Coniferous forest, red-yellow humus–ferrallitic soil | Litter | $0.7 \times 10^6$ | 100 |
| | | Soil (0–20 cm) | $1.6 \times 10^6$ | 400 |
| | Mixed forest, mountainous red-yellow humus–ferrallitic soil | Litter | $0.7 \times 10^5$ | 800 |
| | | Soil (0–20 cm) | $1.5 \times 10^5$ | 300 |
| Kon Plong | Tall-stemmed mixed forest, mountainous red-yellow humus–ferrallitic soil | Litter | $3.9 \times 10^6$ | 400 |
| | | Soil (0–20 cm) | $1.8 \times 10^6$ | 240 |
| Kon Ka Kinh National Park | Eastern part of the park, tall-stemmed forest on a slope, brown tropical thin sandy loam | Litter | $6.1 \times 10^6$ | 126 |
| | | Soil (0–5 cm) | $5.2 \times 10^6$ | 170 |
| | | Soil (5–20 cm) | $6.1 \times 10^6$ | n.d. |
| | Eastern part of the park, tall-stemmed forest, suspended soil | Soil from epiphyte baskets | $2.1 \times 10^6$ | n.d. |
| | Eastern part of the park, tall-stemmed forest on the ridge, mountainous red-yellow humus–ferrallitic soil | Litter | $4.0 \times 10^5$ | 101 |
| | | Soil (0–5 cm) | $5.3 \times 10^4$ | 50 |
| | | Soil (5–20 cm) | $2.0 \times 10^4$ | 75 |
| | Valley tall-stemmed forest, hydromorphic dark humus–ferrallitic soil | Litter | $3.5 \times 10^6$ | 155 |
| | | Soil (0–20 cm) | $1.9 \times 10^6$ | 160 |
| | Mid-mountain tall-stemmed forest, mountainous red-yellow humus–ferrallitic soil | Litter | $6.9 \times 10^6$ | 165 |
| | | Soil (0–20 cm) | $0.6 \times 10^5$ | 160 |
| | Southwestern part of the park, low-stemmed forest, alluvial brown sandy loamy soil | Litter | $3.3 \times 10^5$ | 332 |
| | | Soil (0–5 cm) | $1.4 \times 10^5$ | 252 |
| | | Soil (5–20 cm) | $8.7 \times 10^4$ | 503 |
| | Southwestern part of the park, mixed forest, brown tropical thin clay soil | Litter | $3.3 \times 10^5$ | 40 |
| | | Soil (0–5 cm) | $8.0 \times 10^4$ | n.d. |
| | | Soil (5–20 cm) | $2.0 \times 10^5$ | n.d. |

*3.2. Ecophysiological Properties of Actinomycete Strains Isolated from Tropical Forest Soils*

From the studied natural substrates, we isolated 80 strains of streptomycetes into a pure culture by successive subculturing (9 from the substrates collected in Xuan Son, 20 from Pu Hoat, 7 from Pu Mat, 14 from Kon Chu Rang, 13 from Kon Plong, 17 from Kon Ka Kinh). All strains were tested for antibiotic activity against test organisms belonging to different phylogenetic groups: *Bacillus subtilis*, *Aspergillus niger* and *Candida albicans*. Fifty strains demonstrated significant antibacterial activity. In contrast, only 7 strains showed inhibition of growth of *A. niger*, and 14 strains showed inhibition of growth of *C. Albicans*. The strains that had the highest antibiotic activity were isolated from the soil, litter and suspended soil of the Pu Hoat Nature Reserve. They were tested for a wider range of Gram-positive and Gram-negative test bacteria (*Rhodococcus erythropolis*, *Cellulomonas* sp., *Myxococcus* sp., *Brevibacterium* sp., *Arthrobacter globisporum*, *Promicromonospora* sp.) and showed large growth inhibition zones for most of them [23]. Among the most active were strains represented by the species *Streptomyces cellulosae* and *S. pratensis*, the species identity of which was validated by molecular biological methods. The sequences of the 16S rRNA gene of pure cultures were deposited to GenBank under the accession numbers OP035963 (*Streptomyces cellulosae* strain PHT-ep/21-6) and OP035972 (*Streptomyces pratensis* strain PHT-ep/28-9).

Pure cultures of the isolated actinomycetes were tested for their ability to degrade cellulose and 35% of the strains showed profuse growth on the Hutchinson's medium with filter paper, 19% had medium growth and 38% had poor growth.

Due to the fact that the studied substrates of soils and plant litter had an acidic reaction of the medium, the isolated strains of actinomycetes were tested for the ability to grow at low pH values of the medium. Pure cultures were incubated on media with pH 5 and pH 4. All strains showed growth on a medium with pH 5, and 73% of them grew on a medium with pH 4. An increase in the acidity of the nutrient medium was accompanied by a change in the color of the aerial and substrate mycelium of the incubated strains.

For five strains isolated from soil samples collected from a depth of 20 and 5 cm, as well as from litter and suspended soils, we assessed the temperature range of growth. A strain isolated from soil sampled from a depth of 20 cm showed a maximum growth in the temperature range of 43–45 °C. For actinomycetes isolated from suspended soils and plant litter, the most profuse growth was observed at 30–32 °C. All strains grew in a wide temperature range from 5 to 49 °C.

*3.3. Characterization of Prokaryotic Community Diversity Using High-Throughput Pyrosequencing of the 16S rRNA Gene*

We used samples of alluvial brown soil, leaf litter and suspended soil from epiphyte baskets collected from the territory of the Pu Hoat Nature Reserve for high-throughput pyrosequencing to determine the diversity of the microbial community. The samples from Pu Hoat Nature Reserve have full vertical stratification: suspended soil, litter, soil. Experimental data showed that Pu Hoat isolates are potential biotechnologically important strains: they have high antibiotic and hydrolytic activity and adaptations to high temperatures and low pH.

The total DNA was isolated from samples. The total DNA was used to sequence the V3–V4 region of the 16S rRNA gene. The resulting libraries of 16S rRNA gene fragments were analyzed using the SILVA online resource. Unidentified sequences were excluded from the analysis. The remaining total numbers of reads for libraries derived from alluvial brown soil, leaf litter and soil from epiphyte baskets comprised 1471, 1286 and 1887, respectively, from which 1196, 1214 and 1569 OTUs were generated with a sequence identity level of ≥97%. Diversity indices differed for the libraries derived from the samples (Table 3). The values of the Chao1 index indicating species diversity were lower in the alluvial brown soil sample. The sequences of the 16S rRNA gene of microbial communities were deposited to GenBank under the accession number PRJNA861932.

**Table 3.** Statistical results of sequencing of the V3–V4 region of the 16S rRNA gene.

| Library Name | Number of Reads | Number of OTUs | Goods Coverage, % | Chao1 Index | Simpson Index | Shannon Diversity Index |
|---|---|---|---|---|---|---|
| Alluvial brown soil (B1) | 1471 | 1196 | 30 | 6086.2 | 0.02 | 6.96 |
| Litter (B2) | 1286 | 1214 | 10 | 17191.6 | 0.01 | 7.07 |
| Soil from epiphyte baskets (B3) | 1887 | 1569 | 28 | 8526.4 | 0.01 | 7.27 |

Figure 4 shows the phylogenetic diversity of prokaryotes at the phylum level in the studied samples of natural substrates collected in the territory of the Pu Hoat Nature Reserve. The proportion of *Actinobacteriota* phylum sequences in the three resulting libraries ranged from 7% to 11% of the total number of prokaryotic sequences (Figure 4).

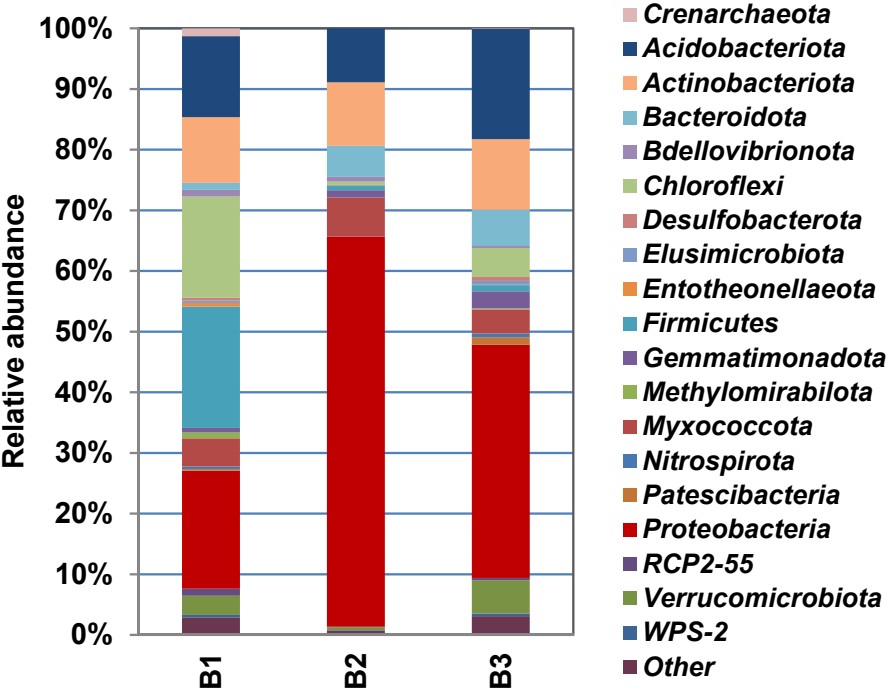

**Figure 4.** Taxonomic classification of prokaryotic communities (at the phylum level) in libraries of 16S rRNA gene fragments derived from samples of alluvial brown soil (B1), leaf litter (B2) and soil from epiphyte baskets (B3) collected in the territory of the Pu Hoat Nature Reserve.

In all studied samples, the phylum *Actinobacteriota* was represented by three classes: *Actinomycetia*, *Thermoleophilia* and *Acidimicrobiia*.

In alluvial brown soil, the phylum *Actinobacteriota* comprised 9% of the total number of prokaryotic sequences. Among the taxa, the families *Acidothermaceae* (28% of the *Actinobacteriota* phylum), *Micromonosporacea* (12%) and *Streptomycetaceae* (11%) were predominant (Table 4).

The *Acidothermaceae* family is predominant in the actinobacterial complex; these are acidophilic thermophilic microorganisms that have thermostable enzymes capable of degrading cellulose. Currently, it is represented by the only genus, *Acidothermus* [36].

**Table 4.** Representation of taxa of the phylum *Actinobacteriota* in alluvial brown soil.

| Class | Order | Family | Content, % |
|---|---|---|---|
| *Actinomycetia* | *Frankiales* | *Acidothermaceae* | 2 |
| | | *Frankiaceae* | 0.1 |
| | | *Sporichthyaceae* | 0.05 |
| | *Corynebacteriales* | *Mycobacteriaceae* | 0.5 |
| | | *Nocardiaceae* | 0.2 |
| | | *Corynebacteriaceae* | 0.01 |
| | *Micromonosporales* | *Micromonosporaceae* | 0.7 |
| | *Streptomycetales* | *Streptomycetaceae* | 0.6 |
| | *Streptosporangiales* | *Streptosporangiaceae* | 0.2 |
| | | *Thermomonosporaceae* | 0.2 |
| | *Pseudonocardiales* | *Pseudonocardiaceae* | 0.2 |
| | *Micrococcales* | *Micrococcaceae* | 0.1 |
| | | *Intrasporangiaceae* | 0.07 |
| | *Propionibacteriales* | *Nocardioidaceae* | 0.04 |
| | | *Propionibacteriaceae* | 0.03 |
| | *Catenulisporales* | *Actinospicaceae* | 0.01 |
| *Thermoleophilia* | *Solirubrobacterales* | *Solirubrobacteraceae* | 0.02 |
| | | TM146 | 0.9 |
| | | YNPFFP1 | 0.2 |
| | | Elev-16S-1332 | 0.03 |
| | *Gaiellales* | *Gaiellaceae* | 0.03 |
| | | uncultured | 0.8 |
| *Acidimicrobiia* | *Acidimicrobiales* | *Acidimicrobiaceae* | 0.09 |
| | | uncultured | 1 |

In plant litter, the phylum *Actinobacteriota* comprised 7% of the total number of prokaryotic sequences. Among the taxa, the families *Micromonosporaceae* (13% of the *Actinobacteriota* phylum), *Microbacteriaceae* (13%) and *Streptomycetaceae* (9%) were predominant (Table 5). According to the literature data, the *Microbacteriaceae* family is most abundant in the litter and phyllosphere; among its representatives, there are plant pathogens [37]. A significant number of identified sequences of the 16S rRNA gene were attributed to bacteria of several genera *Mycobacterium* (10.6%), *Actinoplanes* (8.5%) and *Jatrophihabitans*, *Nocardioides*, *Kineosporia* and *Streptomyces* (6.4% each). Representatives of these genera have been isolated from soil and plant substrates and are most common in the litter and phyllosphere [38–44]. Bacteria of the genus *Mycobacterium* are characterized by acid resistance [45].

The phylum *Actinobacteriota* comprised 11% of the total number of prokaryotic sequences in the soil from epiphyte baskets. Representatives of the families *Acidothermaceae* (24% of the *Actinobacteriota* phylum, 47% of the *Bacteria* domain), *Frankiaceae* (6%) and *Thermomonosporaceae* (5%) were predominant (Table 6). The analysis data indicate the presence of acidophilic and thermophilic actinomycetes in the substrates.

**Table 5.** Representation of taxa of the phylum *Actinobacteriota* in plant litter.

| Class | Order | Family | Content, % |
|---|---|---|---|
| *Actinomycetia* | *Frankiales* | *Acidothermaceae* | 0.2 |
| | | *Frankiaceae* | 0.4 |
| | | *Cryptosporangiaceae* | 0.09 |
| | | *Sporichthyaceae* | 0.05 |
| | *Kineosporiales* | *Kineosporiaceae* | 0.6 |
| | *Corynebacteriales* | *Mycobacteriaceae* | 0.5 |
| | | *Nocardiaceae* | 0.05 |
| | *Micromonosporales* | *Micromonosporaceae* | 0.9 |
| | *Streptomycetales* | *Streptomycetaceae* | 0.7 |
| | *Micrococcales* | *Microbacteriaceae* | 0.9 |
| | | *Cellulomonadaceae* | 0.4 |
| | | *Promicromonosporaceae* | 0.1 |
| | *Propionibacteriales* | *Nocardioidaceae* | 0.2 |
| *Thermoleophilia* | *Solirubrobacterales* | *Solirubrobacteraceae* | 0.05 |
| | | *Conexibacteraceae* | 0.1 |
| | | *Patulibacteraceae* | 0.05 |
| | | TM146 | 0.2 |
| | | uncultured | 0.05 |
| | | Elev-16S-1332 | 0.3 |
| | *Gaiellales* | *Gaiellaceae* | 0.05 |
| *Acidimicrobiia* | *Acidimicrobiales* | *Acidimicrobiaceae* | 0.4 |
| | | *Iamiaceae* | 0.1 |
| | | uncultured | 0.2 |
| | | *AcidimicrobialesIncertae Sedis* | 0.07 |

**Table 6.** Representation of taxa of the phylum *Actinobacteriota* in soil from epiphyte baskets.

| Class | Order | Family | Content, % |
|---|---|---|---|
| *Actinomycetia* | *Frankiales* | *Acidothermaceae* | 3 |
| | | *Frankiaceae* | 0.7 |
| | | uncultured | 0.2 |
| | | *Sporichthyaceae* | 0.1 |
| | *Streptosporangiales* | *Thermomonosporaceae* | 0.5 |
| | | *Streptosporangiaceae* | 0.08 |
| | *Kineosporiales* | *Kineosporiaceae* | 0.2 |
| | *Corynebacteriales* | *Mycobacteriaceae* | 0.04 |
| | | *Corynebacteriaceae* | 0.04 |
| | *Micromonosporales* | *Micromonosporaceae* | 0.5 |
| | *Pseudonocardiales* | *Pseudonocardiaceae* | 0.4 |
| | *Streptomycetales* | *Streptomycetaceae* | 0.3 |
| | *Micrococcales* | *Microbacteriaceae* | 0.05 |
| | *Propionibacteriales* | *Nocardioidaceae* | 0.5 |

**Table 6.** *Cont.*

| Class | Order | Family | Content, % |
|---|---|---|---|
| *Thermoleophilia* | *Solirubrobacterales* | *Solirubrobacteraceae* | 0.09 |
| | | *Patulibacteraceae* | 0.1 |
| | | TM146 | 0.5 |
| | | YNPFFP1 | 0.1 |
| | | Gsoil-1167 | 0.02 |
| | | Elev-16S-1332 | 0.7 |
| | *Gaiellales* | *Gaiellaceae* | 0.2 |
| | | uncultured | 1 |
| *Acidimicrobiia* | *Acidimicrobiales* | *Acidimicrobiaceae* | 0.3 |
| | | *Iamiaceae* | 0.09 |
| | | uncultured | 1 |

Figure 5 shows a thermal map of the biodiversity of representatives of the phylum *Actinobacteriota* at the genus level in the studied communities of alluvial brown soil, leaf litter and suspended soil from epiphyte baskets. In the library of 16S rRNA gene fragments from an alluvial brown soil sample, predominant were *Acidothermus*, uncultivated *Gaiellales* and *Acidimicrobiia* (17.1%, 13.9% and 9.5% of the total number of sequences, respectively). The same dominants were noted for a sample of soil from epiphyte baskets (15.6%, 11.5% and 7.8%, respectively). In the leaf litter library, a significant number of identified 16S rRNA gene sequences were attributed to bacteria of the genera *Solirobrobacterales* (9.0%), *Streptomyces* (6.0%), *Actinoplanes* (6.0%), *Acidothermus* (5.3%) and *Jatrophihabitans* (5.3%) (Figure 5).

Comparison of the OTU libraries of the 16S rRNA genes of the phylum *Actinobacteriota* using a Venn diagram revealed that the sample of alluvial brown soil contained a greater number of unique sequences than the samples of leaf litter and soil from epiphyte baskets (Figure 6).

Using the Local Mapper module of the iVikodak program and the KEGG database, we conducted an analysis of the enzymes of the starch and sucrose metabolism pathway that are potentially produced by the studied communities (Figure 7).

The enzyme cellulase [EC 3.2.1.4] which degrades cellulose to D-glucose was identified in all three analyzed natural substrates collected from the Pu Hoat Nature Reserve. The enzyme was most abundant in samples of alluvial brown soil and soil from epiphyte baskets and to a lesser extent in a sample of leaf litter (Figure 7). The contribution of individual taxa of the *Actinobacteriota* phylum to the cycle of starch and sucrose degradation is shown in Figure 8. Of note is the high activity in the biodegradation of cellulose shown by representatives of the genus *Acidothermus* (21.53% and 22.61% for alluvial brown soil and soil from epiphyte baskets, respectively). In leaf litter, predominant in the degradation of cellulose are representatives of the genera *Actinoplanes* (24.65%) and *Streptomyces* (19.98%).

These results confirm the feasibility of the degradation of cellulose to glucose by bacteria of the *Actinobacteriota* phylum from all studied samples of natural substrates collected in the Pu Hoat Nature Reserve. Thus, the metabolic potential of the soil can be predicted by analyzing the species composition and metabolic potential of prokaryotic communities.

The use of molecular biological research methods made it possible to significantly expand the data on the diversity of the actinomycete complex in the studied samples of natural substrates and obtain information on taxa that are difficult or impossible to identify in the soil using the classical method of sowing on solid selective media.

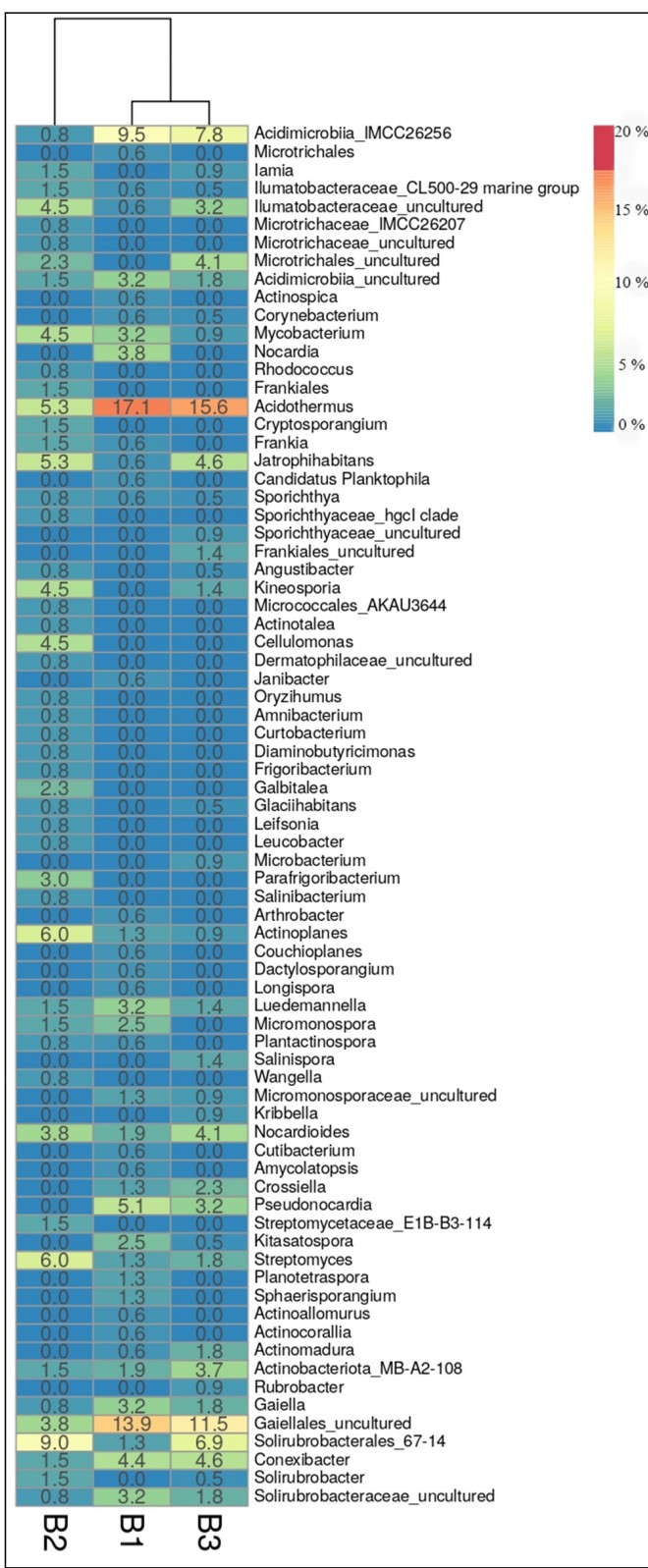

**Figure 5.** Taxonomic classification of genera within the *Actinobacteriota* phylum in libraries of 16S rRNA gene fragments (according to SILVA) derived from samples of alluvial brown soil (B1), leaf litter (B2) and soil from epiphyte baskets (B3) collected from the territory of the Pu Hoat Nature Reserve. The numbers in the diagram indicate the percentage of the number of sequences in the library within the *Actinobacteriota* phylum in each sample examined.

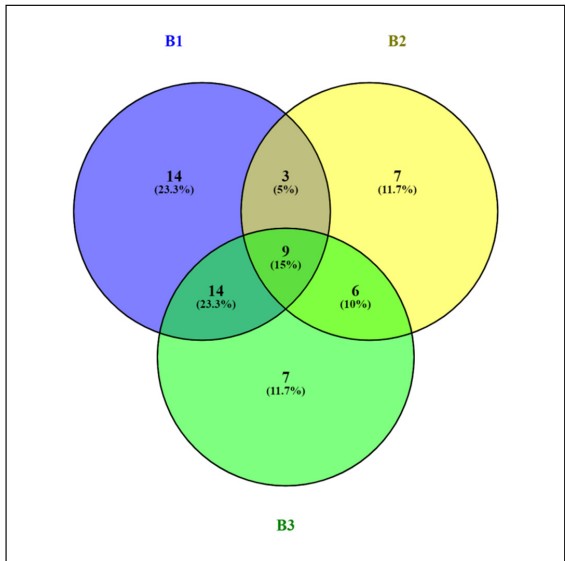

**Figure 6.** Venn diagram showing the logical relation between OTU 16S rRNA genes of the bacterial community of the *Actinobacteriota* phylum in samples of alluvial brown soil (B1), leaf litter (B2) and soil from epiphyte baskets (B3) collected in the Pu Hoat Nature Reserve.

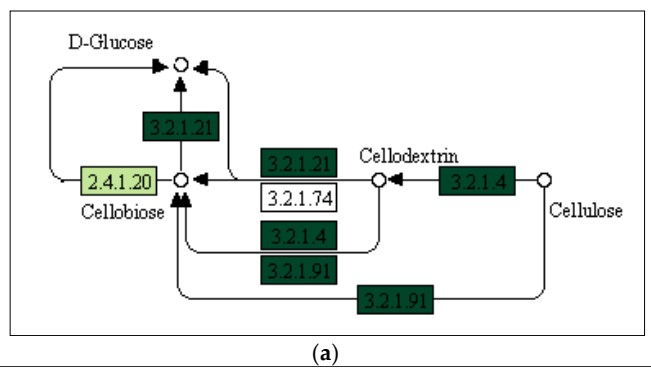

(**a**)

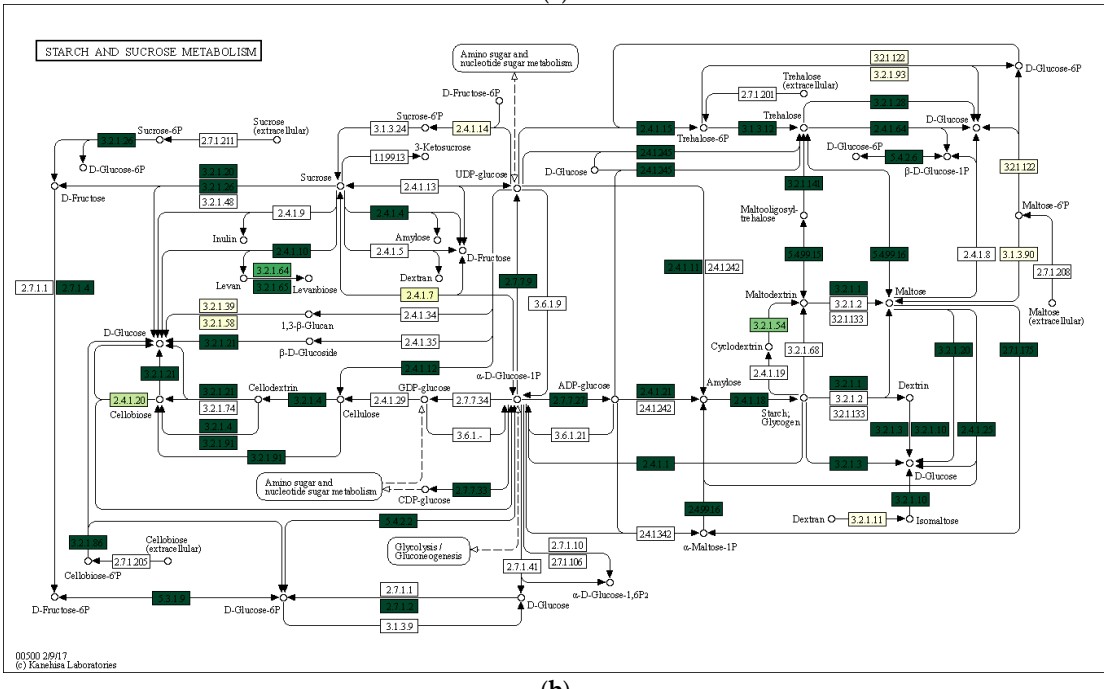

(**b**)

**Figure 7.** *Cont.*

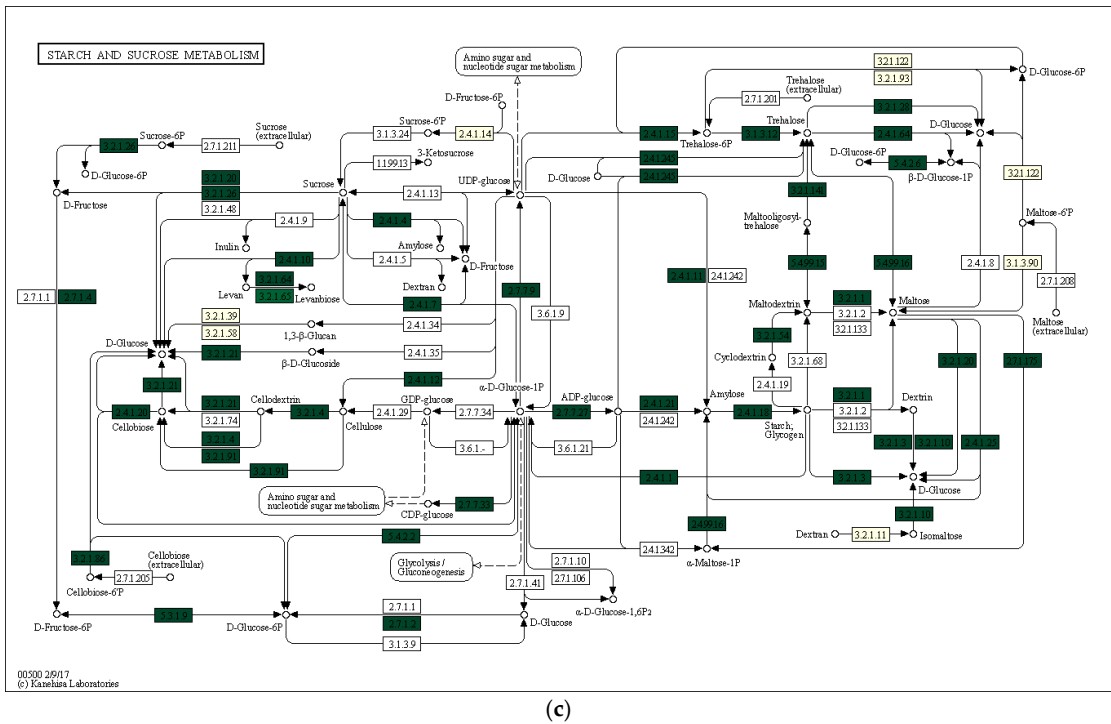

(**c**)

**Figure 7.** Key degradation enzymes of starch and sucrose metabolism in samples of alluvial brown soil (**a**), leaf litter (**b**) and soil from epiphyte baskets (**c**) collected in the Pu Hoat Nature Reserve. The saturation of the green color is determined by the representation of a particular enzyme in the genomes of microorganisms. EC numbers are taken from the enzyme nomenclature database.

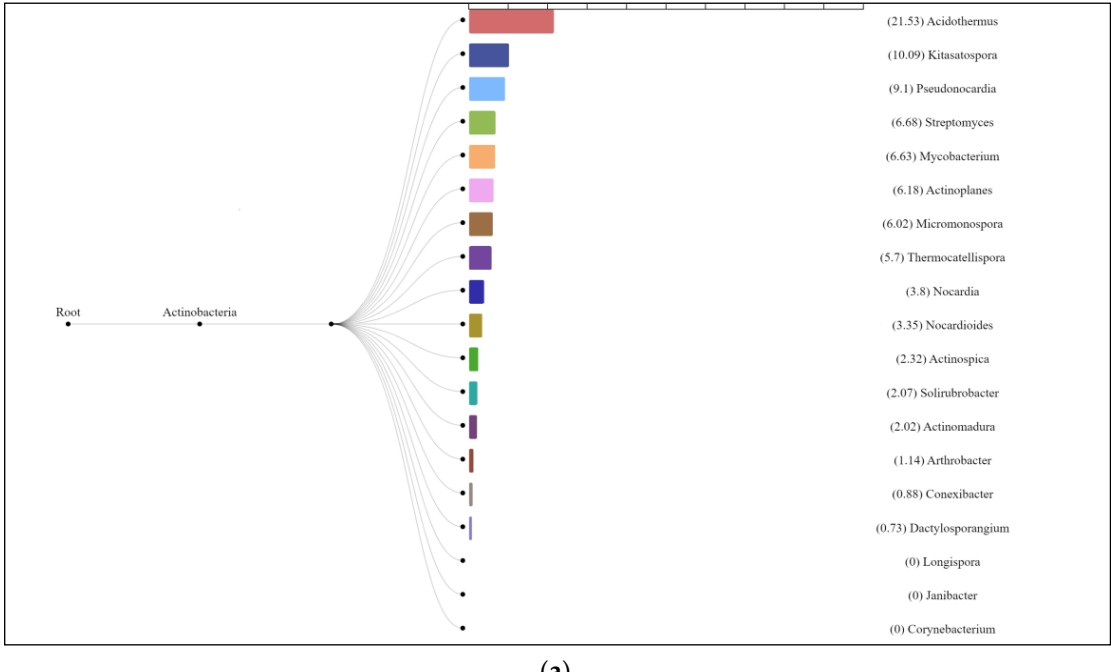

(**a**)

**Figure 8.** *Cont.*

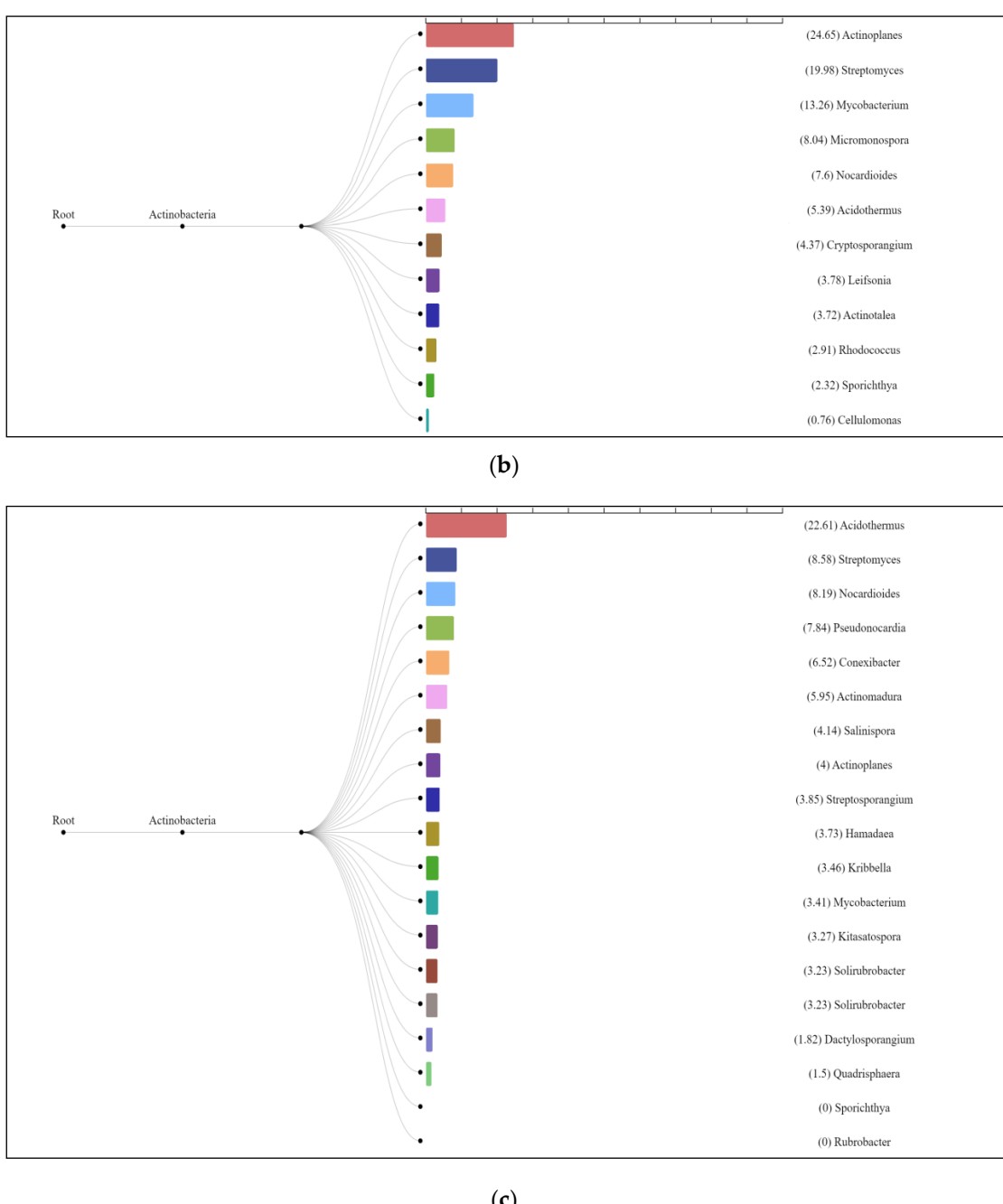

**Figure 8.** Dendrograms showing the contribution of bacteria of the *Actinobacteriota* phylum to the degradation cycle of starch and sucrose metabolism in samples of alluvial brown soil (**a**), leaf litter (**b**) and soil from epiphyte baskets (**c**) collected in the Pu Hoat Nature Reserve.

## 4. Discussion

A characteristic feature of tropical forests is the high rate of biological cycle of substances. High rates of decomposition and mineralization of organic matter are primarily the result of intensive activity of microorganisms. The studies of the substrates of the tropical forests of Vietnam showed that actinomycetes are an integral component of all microbial complexes in the protected zones of Vietnam. High values of abundance (up to $10^8$ CFU/g), length of actinomycete mycelium (up to 1000 m/g) and cellulolytic activity indicate their important role in the destruction of organic matter in the studied substrates. The obtained values are comparable with the content of actinomycetes in the ferrallitic soils of the tropical forests of Brazil ($10^5$–$10^6$ CFU/g) [19].

The taxonomic diversity of actinobacteria varies in different substrates. The use of the classical inoculation method showed that the richest diversity is found in the Kon Ka Kinh National Park. It is obvious that the differences in the diversity of mycelial actinobacteria can be explained by the differences in soil pH and the richness of the plant community. According to the results of high-throughput pyrosequencing of the 16S rRNA gene, suspended soils forming in epiphyte baskets which are unique tropical forest substrates have a wide variety of taxa of different levels (1285 species). Similar values (1339 species) were observed in soil samples, while in plant litter, the taxonomic diversity is several times lower: 297 species. The limited distribution of actinomycetes in plant litter is due to competitive interactions that limit the germination of spores in this substrate, which confirms the role of actinobacteria in the mineralization of organic matter, the ecological strategy of which is based on the consumption of hard-to-get organic compounds. The results obtained for the substrates of the Pu Hoat Nature Reserve exceed the data on the taxonomic diversity of similar ecotopes in the Xishuangbanna tropical forest of China [46].

Bioinformatics analysis of the obtained data revealed a high proportion of the *Acidothermaceae* family in the actinobacterial complex; experimental data showed that their hydrolytic enzymes work well under conditions of an acidic reaction of the medium and high temperatures. Actinomycetes of the *Streptomyces* genus are also known to be producers of cellulolytic enzymes [47]. However, most sources report that streptomycete cellulases are active in a weakly alkaline medium [48]. However, testing showed that the isolated strains are acidophilic and acid-tolerant organisms capable of degrading cellulose. In 2018, based on many studies, the possibility of using acidophilic and acid-tolerant actinomycetes for sustainable agriculture on acidic soils was suggested, due to the fact that they are capable of synthesizing phytohormones, cytokinins and gibberellins necessary for the normal growth and development of cultivated plant crops [49].

A number of studies showed that actinomycete complexes include thermotolerant and thermophilic strains. Such organisms may be of interest as producers of active hydrolytic enzymes [50].

From suspended soils, which are a special type of organo-mineral substrate formed by epiphytes, we isolated two strains of streptomycetes with promising biotechnological properties: *Streptomyces cellulosae* strain PHT-ep/21-6 and *Streptomyces pratensis* strain PHT-ep/28-9. These organisms were capable of degrading cellulose, had a high antimicrobial activity against test cultures of different phylogenetic groups and were acidophilic and thermophilic.

Rainforest actinobacteria are distinguished by a unique combination of ecophysiological properties: adaptations to low pH values, a wide range of temperatures and altering dry and wet climatic seasons. The combined presence of these characteristics determines the high biological activity of actinomycete complexes. In addition to the destruction of organic compounds and enzymatic activity, actinomycetes of these habitats are of interest as producers of antibiotics. It can be assumed that in the conditions of high rates of biological cycle and intensive migration of antibiotic resistance genes among bacteria, actinomycetes must maintain the synthesis of antibiotic compounds at a certain level. Some properties of actinomycetes, such as the ability to colonize the surface of plants or secrete antibiotic substances against phytopathogens, the synthesis of specific extracellular proteins and the degradation of phytotoxins, indicate their participation in biological control and stimulation of plant growth [51]. This determines the important ecological role of actinomycetes in the stability of tropical ecosystems, namely in plant protection from phytopathogens, formation of soil resistome and soil self-purification. Producing acid-resistant and thermostable enzymes and antibiotics is of great importance for biotechnology and agriculture.

The data obtained confirm the importance of actinomycetes in the destruction of organic matter in acidic soils of tropical regions.

## 5. Conclusions

We established that mycelial actinobacteria are an integral component of prokaryotic communities in the protected zones of Vietnam. The high abundance, great length of the actinomycete mycelium and wide taxonomic diversity indicate that actinomycetes occupy a significant place in the biological cycle of tropical forests. The highest abundance of actinomycetes ($10^8$ CFU/g) and the greatest mycelium length (1000 m/g) were recorded in the litter of the broadleaf forest in the Xuan Son National Park located in the northern part of Vietnam. In the southern part of Vietnam, in the substrates of the Kon Ka Kinh National Park, we observed the smallest length of actinomycete mycelium, but at the same time, the richest taxonomic diversity.

Suspended soils formed in epiphyte baskets are of great interest for further research. The results of high-throughput sequencing showed a high taxonomic diversity and the presence of rare actinomycete genera in such substrates of the Pu Hoat Nature Reserve. The strains of streptomycetes isolated from them had the highest antibiotic activity.

Bioinformatics analysis and experimental data showed that the soils and associated plant substrates of the tropical forests of Vietnam are the habitat of potential biotechnologically important strains. In particular, some representatives of the phylum *Actinobacteriota* have a high biological activity at low pH and high temperatures.

**Author Contributions:** Conceptualization, T.A.G. and Y.A.D.; methodology, T.A.G. and T.L.B.; software, T.L.B., D.S.S. and L.V.L.; validation, T.A.G., T.L.B. and D.S.S.; formal analysis, Y.A.D., T.L.B. and D.S.S.; investigation, Y.A.D.; resources, A.V.A. and G.T.H.P.; data curation, T.A.G. and T.L.B.; writing—original draft preparation, T.A.G.; writing—review and editing, Y.A.D., T.L.B. and D.S.S.; visualization, Y.A.D. and D.S.S.; supervision, T.A.G.; project administration, N.A.M. and A.V.G.; funding acquisition, T.A.G., T.L.B., D.S.S., A.V.A., N.A.M. and A.V.G. All authors have read and agreed to the published version of the manuscript.

**Funding:** The study was supported by the Ministry of Science and Higher Education of the Russian Federation, agreement no. 075-15-2021-1396.

**Data Availability Statement:** Data are contained within the article.

**Conflicts of Interest:** The authors declare no conflict of interest.

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
