# Peer review of "Soil Actinomycetes of Vietnam Tropical Forests"

_forests, doi:10.3390/f13111863_

Round 1

Reviewer 1 Report

The results were very interesting and corroborated with the literature. However, the authors left something to be desired in the discussion of the results, they could have explored the discussion further, emphasizing the importance of this study;

Work the discussion further.

Author Response

Thank you for your comments.  We have carefully reviewed the comments and have revised the manuscript. Changes to the manuscript are shown in color.

We hope the revised version is now suitable for publication and look forward to hearing from you in due course.

Reviewer 2 Report

Sorry, there is a lot here, but it is what is not here that lets you down. I'd be prepared to consider submitting this as two papers. One on the samples and their differences between parks and one on the sequencing and subseuqent inferred bioinformatic modelling.

Abstract What is the point of this work? The presentation of the abstract is clumsy and suggests a series of only moderately related experiments. I would suggest the authors rethink the abstract and apply an approach in keeping with an approach such as the format or other approach. This may give them some guidance: https://services.unimelb.edu.au/__data/assets/pdf_file/0007/471274/Writing_an_Abstract_Update_051112.pdf

The molecular work appears to be from just one location of the several mentioned, so perhaps the paper should focus on that area and use the others as comparisons for the other experiments or not at all?

Introduction

The written English is much better in this section, but the presentation and content both need work.

Ln 41 species diversity of what? Richest according to who? Reference needed.

Ln 41-46 are a bit too repetitive and detract from the key messages of understanding and protection. Needs to be re-written.

Ln 50-54 would be nice to see a reference to this exceptional distribution.

Ln 62-66 so is it a neutral or an acidic pH that is important. This paragraph is presumptive and currently presents slightly conflicting arguments and should be rethought and rewritten.

Ln 65-77 the ‘resistome’ and the antibiotic importance of actinomycetes could be combined into a single paragraph, with the importance at the start.

Hold the phone. The introduction finishes on the importance of antibiotic resistance and horizontal gene transfer as a function of temperature. What is this paper going to be about? Neither of these two things from what the abstract claimed.

Material and methods.

Ln 90-94 is there a seasonality to the collection or some form of temporal measure?

The park details are great and relatively consistent, but how do you sample parks that are thousands of ha in size to get comparable samples? I can’t see how the metadata is going to be used in any comparison at the moment beyond this one differs from that. Is that the point?

How are the isolates recovered from soil and leaf material? You go straight to the media without reference to the recovery dilutions or the solutions used. How much replication was used (should be noted in legend of table 2).

Ln 227-229 I would like to see more on a staining technique only currently in a lab book in Russian. Acridine orange is a DNA stain, so how is it being used to only determine the length of actinomycete mycelia?

Ln 248 which solid media was used?

So all isolates were sequenced, but it then looks like all soils were used for 16s rRNA analysis, but the abstract implied this was only for one soil. Which is it? What’s actually been done?

Results

Start with pH. When and how was that determined?

What is figure 3? The legend really does nothing to assist with what is being reported and there is no Y-axis label. Might want to check the taxonomic labels as well, given it is impossible to tell what level of taxonomic diversity is being reported and from what.

Ln 315-316 so there are at least two samples from this park. Reads like material and methods doesn’t it?

Ln 364 epiphytes and soil are not terms you see together very often. In fact, this might be the first time I have ever.

Ln 378 Can I suggest that this might make for a second paper? It has little bearing or comparison to what else is being reported.

Ln 456 these proposed metabolomics pathways from the sequence data should have been proposed in the material and methods. They are hypothetical, but presented as fact. Another argument for this being in a second paper.

Figure 7 legend. What are the different colour boxes? What are the numbers representative of? These legends all need work.

Discussion

Ln 520 add in the cellulosic activity and I will give you that. Without it what it is evidence of is high numbers and (if the acridine orange method is any good) a lot of potential coverage.

Ln 540 you did not get the enzyme activity data from the bioinformatics. It is inferred from the analysis.

Ln 587 you could define more clearly what this is that would be good. My belief is it is simply a form of detritus rather than soils, but perhaps there is sand, silt and clay in there……although how would be interesting to speculate.

Round 2

Reviewer 2 Report

Thank you for taking the time to revise your manuscript and to go into some detail to explain your rationale and methods to me in your response. Sadly though, these are still missing from the paper, which although better is still not ready for publication.

Abstract.

Much better.

Ln 20 “from six of Vietnam’s”

There are still several English issues to resolve.

Ln 21 “Inoculation of Gause” or “inoculation on to Gause”, but not as presented.

Ln 23 “mycelium was as”

Ln 24 to 33 still needs some work. The bioinformatics is just from the Pu Hoat reserve, but the screening was done on the broader recovery, wasn’t it. Or have you included the pure culture identifications in this? The clarity of presentation needs to be improved here. I would present the plate screening and then the bioinformatics from the sub set as proof.

Introduction.

That is so much better. Thank you and well done.

Material and methods

There is still a lack of detail in the table 1 legend. You have the protected area, the sampling location, which appears to be grouped by geography and altitude, and the soil type according to a classification, but whose? Legends need to stand alone with their tables and figures.

Ln 293-294. You wrote me a beautiful and brief summary of the acridine orange method, but why is it not in the paper? Guess where I’d like to see it? Magnification used for measurement? Measurements taken with what? Graticules or software?

Ln 307 – were these stationary liquid cultures? What was the inoculation density? So much more you could tell the reader.

The focus in here is better, but still needs more work.

Results.

Table 2. The repetition of some of the columns is fine, but the legend should tell me how many replicates produced these abundance and length values. Are these means or one off absolutes? More detail is needed!

Figure 3. The legend should state how these taxonomic classifications were identified. What level they are at (al capitalised and non-italicised at present – so not genus and species?) and the axis are both lacking titles. You did a good job on figure 4, so it is obvious you can do this!

Need to check the Latin nomenclature throughout. Looks like a lot of the species have remained capitalised.

Section 3.2 starts with methodology. Why not start with it being an indepth investigation of the Pu Hoat reserve? Again, there was more rational for this in your response, which is not presented in the methods.

Table 4, 5 and 6 need more column headings please. What aspect of the taxa are we looking at and if these are above the order of genus, then italics are not generally needed. Try to fit the names onto a single column. Can you also please add more information to the legends so that these tables can stand alone without the text.

Figure 5 is a heat map, but nowhere in the legend is the magnitude of similarity of the colour gradient representation explained. This should be there.

Ln 530. The Venn diagram is the method of presentation, not the method of comparison. Pedantic perhaps, but correct. Please amend the text to highlight the method by which the difference is actually assessed and not simply presented. Good legend for the figure 6 though.

Figure 7 legend should reflect that these are predicted metabolic pathways from the bioinformatics. I also wonder of the numbers (what are these, the legend does not say?) in the green boxes will be visible when published. Are the white, yellow, light green and green boxes important? If so, what do they signify?

Figure 8 is again based on the bioinformatics or on the plate assays? Is the top bar the % contribution of the distribution? Could the font of the Phylum be made bigger? Given nothing looks like it is above a 25% contributor, do we need so much white space?

Discussion

Again you sent me some lovely crafted responses, but few of them appear to have been incorporated into the manuscript. Why? Some of the observations you wrote about in the response are also not reflected in this discussion, but you argued they were important. The closing line also says they are the “main destructors of organic matter”. Couple of issues with this. You only report on the actinomycetes, so this is an inductive response not a deductive one. Secondly, word choice is poor. Consider, “The data obtained confirm the importance of actinomycetes in the breakdown of organic matter in acidic soils of tropical regions of several of Vietnam’s protected areas.” You could then add something on the significance of this, although that could equally be reflected in the conclusion, which currently does not align with the abstract in its rewritten form.

Author Response

Thank you for your comments.  We have carefully reviewed the comments and have revised the manuscript. Changes to the manuscript have been added. 

We hope the revised version is now suitable for publication and look forward to hearing from you in due course.
